# Association of Low Sputum Smear Positivity among Tuberculosis Patients with Interferon-Gamma Release Assay Outcomes of Close Contacts in Japan

**DOI:** 10.3390/ijerph16193713

**Published:** 2019-10-02

**Authors:** Tsuyoshi Ogata, Natsuki Nagasu, Ritei Uehara, Kunihiko Ito

**Affiliations:** 1Tsuchiura Public Health Center of Ibaraki Prefectural Government, Tsuchiura 300-0812, Japan; 2Mito Public Health Center of Ibaraki Prefectural Government, Mito 300-0852, Japan; n.nagasu@pref.ibaraki.lg.jp; 3Department of Epidemiology for Community Health and Medicine, Kyoto Prefectural University of Medicine, Kyoto 602-8566, Japan; ruehara@koto.kpu-m.ac.jp; 4Japan Anti-tuberculosis Association, Tokyo 101-0061, Japan; keitou78@pd5.so-net.ne.jp

**Keywords:** tuberculosis, smear, AFB, contact infections, IGRA

## Abstract

Risk prediction and response measures may differ in tuberculosis (TB) patients with low sputum smear positivity for acid-fast bacillus (AFB) compared to those who are smear negative. However, previous studies using the tuberculin skin test (TST) did not show that differences in measures are important. This study compared results of interferon-gamma release assays (IGRA) between contacts of pulmonary TB patients with AFB smear positivity and those with smear negativity using QuantiFERON^®^-TB Gold In-Tube (QFT) assays. Close contacts of TB patients with culture-confirmed infections between April 2010 and December 2012 in Ibaraki, Japan, were enrolled, and 439 Japanese contacts of 129 index TB patients were examined. Adjusted odds ratios of QFT in contacts were 0.68 (95% confidence interval: 0.17–2.8) for AFB scanty patients, 1.12 (0.45–2.8) for AFB 1+, 1.20 (0.48–3.0) for AFB 2+, and 4.96 (1.9–12.9) for AFB 3+, compared to those who were smear negative. Differences in IGRA positivity were not significant between close contacts of TB patients with low positive and negative smears.

## 1. Introduction

Tuberculosis (TB) is a major health concern, causing an estimated 1.3 million deaths in 2017 [1]. Contact exposure to *Mycobacterium tuberculosis* (Mtb) from TB patients increases the risk of contracting TB infections [2]. Infection risk resulting from exposure to pulmonary TB patients is influenced by host, bacterial, and environmental factors. 

Sputum smear acid-fast bacillus (AFB) microscopy in index TB patients is used as an important predictive factor for pulmonary TB infection risk among contacts [2,3], as well as for risk-stratification of Mtb exposure. Therefore, sputum smear positivity or negativity for AFB of the index patient occasionally affects response measures of TB. The World Health Organization (WHO) formerly recommended that in low- and middle-income countries, contact investigations must be conducted for households and close contacts when the index case has sputum smear positive pulmonary tuberculosis [4]. In Japan, if a sputum smear positive pulmonary TB patient refuses admission, the local government by law mandates compulsive hospitalization in a designated hospital.

In detail, smears are graded as negative, scanty, 1+, 2+, or 3+ for AFB. The grades of smear positivity are used for risk stratification to implement response measures; for example, to delineate responses between patients with sputum smear negativity and low sputum smear positivity. Therefore, it is important to validate the degree to which low smear positivity for AFB can predict transmission risk to contacts as compared to negative smears.

TB infections remain latent in most infected contact cases. [5] Tuberculin skin tests (TSTs) and interferon-gamma release assays (IGRA) are used for identification of latent tuberculosis infection (LTBI). IGRA detects interferon-gamma release from lymphocytes after in vitro incubation of whole blood with Mtb antigens. According to a review by the WHO, the pooled risk ratio estimate for TST was 1.49 (95% confidence interval: 0.79–2.80), and that for IGRA was 2.03 (1.18–3.50). Although the estimate for IGRA was slightly higher than that for TST, the 95% confidence intervals for the estimates for TST and IGRA overlapped and were imprecise. The WHO guidelines recommend that either TST or IGRA be used to test for LTBI and that the availability and affordability of the tests will determine which will be chosen by clinicians and program managers [5].

A few studies using TSTs compared LTBI in contacts of patients with low smear positivity versus smear negativity [6,7,8,9]. However, these studies did not show any differences between close contacts of patients with low sputum smear positivity versus those with negative smears. Studies that used IGRAs for comparing contacts of patients positive for low grade sputum smear with contacts of patients with negative smears have not been reported.

This study used IGRAs to evaluate infection status among contacts in order to examine the differences in infections between close contacts of pulmonary TB patients with low sputum smear positivity for AFB and those of patients with smear negative results.

## 2. Materials and Methods

### 2.1. Study Design and Setting

This study used a cross-sectional design and recruited patients from hospitals in the Ibaraki prefecture of Japan. The Ibaraki prefecture neighbors metropolitan Tokyo and in 2012 had an annual tuberculosis incidence rate of 14.0 cases per 100,000 population [10].

### 2.2. Index TB Patients and Exposure

Index TB cases eligible for inclusion in this study involved patients with pulmonary TB diagnosed by designated hospitals with confirmed Mtb culture growth that were registered at any government public health center in Ibaraki from April 2010 to December 2012. We randomly enrolled participants from among index patients who had at least one close contact. The public health centers investigated the patients and collected information regarding cough, maximum smear grading for AFB of three consecutive sputum samples, and chest radiograph results. The exposure measured in this study was the AFB grading of three sputum smears in index patients. Low smear positivity was defined as a grade of either scanty or 1+.

### 2.3. Subject Contacts

Subject contacts were considered close contacts of enrolled index TB patients. Per the Japan national TB control program regulations and 2010 guidelines [11], contacts of index TB patients were visited and interviewed by public health nurses. Close contacts included in this study were those who had lived or worked in the same room as the index TB patient. Since the prevalence of tuberculosis and LTBI was shown to be higher in populations from foreign countries or those aged over 60 years old [12,13], contacts that met these criteria were excluded from the study.

### 2.4. Outcome

Since nearly all Japanese residents have received mandatory Bacillus Calmette-Guerin (BCG) vaccines during childhood, the IGRA was more suited for determining LTBI [14]. Furthermore, published guidelines recommend that the IGRA, QuantiFERON^®^-TB Gold In-Tube (QFT) assay (Cellestis Limited, Carnegie, Australia), be preferentially used in health screenings, except among children, two or three months after exposure to TB [11]. The QFT antigens included a mixture of peptides representing ESAT-6, CFP-10, and TB7.7 proteins. In this study, blood samples were collected from contacts by public health nurses and transported with appropriate temperature control to the Ibaraki Health Service Association laboratory. The whole blood was collected in three tubes (specific antigen, positive control, and negative control) at the temperature of 22 ± 5 ℃. It stimulated the antigenic mixture in the tube. The tubes were incubated at 37 ℃ for 16 to 24 h, and centrifugated. INF-γ release was measured by ELISA. QFT assays were performed following the manufacturer’s recommended protocol. The assay was evaluated by the difference of readouts between the antigenic mixture and the negative control.

The main study outcome measured was QFT assay positivity. Cut-off value of 0.35 IU/mL was used in the analysis according to the guidelines [11].

### 2.5. Statistical Analysis

Contact characteristics were summarized. The QFT positivity rates of contacts were calculated and compared across index patients or contact risk factors. Logistic regression analyses were used to calculate adjusted odds ratios. Data were presented as counts with percentages or odds ratios with 95% confidence intervals (95% CI). Statistical analyses were performed using R (version 2.4-0; The R Foundation for Statistical Computing, Vienna, Austria).

### 2.6. Ethical Approval

The study protocol was approved on 19 November 2012 by the Ibaraki Prefecture Epidemiological Research Joint Ethics Review Committee (protocol number: H24-01). The study was conducted in accordance with the Declaration of Helsinki. Per protocol, we collected data from original databases established by individual public health centers following data anonymization. Per protocol, instead of written informed consent, the study plan and means to opt out of the study were advertised on the Ibaraki prefecture website homepage.

## 3. Results

A total of 129 index TB patients registered at public health centers in the Ibaraki prefecture during the study period were randomly selected, and a total of 515 close contacts of these patients screened were included in this study. Of these, 43 foreign contacts and 33 Japanese contacts aged 60 years and over were excluded and the remaining 439 close contacts were included in the analysis (Figure 1). The mean and medium (range) of number of contacts per TB case were, respectively, 4.8 and 2 (1–43).

Table 1 describes the characteristics of contacts based on the index patient’s AFB smear status, with 103, 41, 115, 132, and 48 contacts who had negative, scanty, 1+, 2+, and 3+ grades, respectively.

A total of 57 (13%) contacts tested positive using QFT analysis. Four contacts with QFT-positive results contracted TB. Crude odds ratios of contacts showed higher QFT-positive yield in contacts of patients with cough, and high sputum smear grading (trend *p*-value = 0.0016). However, they did not show higher yield in contacts of patients with cavity or older age. In the multivariate analysis, the adjusted odds ratios of contact QFT based on the index patient’s smear status compared to that of smear negative patients were 0.68 (95% CI: 0.17–2.8) for scanty AFB smears, 1.12 (95% CI: 0.45–2.8) for AFB 1+ smears, 1.20 (95% CI: 0.48–3.0) for AFB 2+ smears, and 4.96 (95% CI: 1.9–12.9) for AFB 3+ smears (Table 2).

## 4. Discussion

The study findings showed that close contacts of patients with sputum smears graded AFB 3+ had higher IGRA positivity than those of patients with smear negative results. We did not find significant differences in IGRA positivity between close contacts of patients with low and negative sputum smear AFB grades. As a result, this study did not provide any evidence indicating differences in transmission risk between contacts of patients with low sputum smear positivity and smear negativity.

In previous studies that did not classify patients based on sputum smear grading, contacts of patients with sputum smear positivity showed higher LTBI positivity rates than those of patients with smear negativity [2,15,16,17,18,19,20]. However, since sputum smears were not graded, these studies were unable to directly compare LTBI positivity between contacts of patients with sputum smear AFB 1+ and those of patients with smear negativity. Other studies conducted in high-incidence countries reported that contacts of patients with high sputum smear positivity had higher TST or IGRA yields than those of patients with low smear positivity. However, these studies did not include contacts of patients with negative sputum smears [21,22,23,24,25].

Several studies compared infection status among close contacts of patients with sputum smear gradings for AFB, including patients with smear negativity, using TST [6,7,8,9]. In these studies, there was a significant trend for increases in the number of infections in contacts as the degree of smear positivity in index TB patients increased. However, these studies failed to show the differences in infectivity between contacts of patients with low grade sputum smear positivity and sputum smear negativity. In addition, previous studies have not used IGRA measures to compare infections in contacts of patients with low grades and negative AFB smears. Our results and the lack of direct findings in other studies indicate that there is insufficient evidence to suggest differences in transmission risk or recommended response measures between low sputum smear positivity and smear negativity.

We used evaluations of IGRA outcomes in this study. Nienhaus et al. reported that agreement of IGRA and TST is excellent with little potential that TST is more likely to detect old infections than IGRA [10]. A systematic review and meta-analysis found that commercial IGRAs had a higher positive and negative predictive value for progression from latent infection to active disease [26]. Both TST and IGRA are widely available and have been used for LTBI testing, and the advantages and disadvantages of these tests have been previously described [27]. Japanese guidelines recommend that IGRA should be preferably used for health screening among adults because of higher specificity in BCG-vaccinated individuals [11,14]. However, the WHO guidelines note that BCG vaccination should not be a determining factor when selecting a test [5]. The present study used IGRA testing and did not show evidence warranting the implementation of different infection prevention measures for close contacts of patients with low sputum smear positivity and those of patients with smear negative results.

The 2012 and 2013 WHO recommendations stated that contact investigations should always be done when the index case has sputum smear positive pulmonary TB [4,28], whereas 2015 recommendations targeting high- or upper middle-income countries and the 2018 guidelines do not mention smear positivity results for initiation of contact investigations [5,29]. In the United States, the Centers for Disease Control and Prevention noted that index patients with positive AFB sputum smear results or pulmonary cavities should receive the highest priority for examination, and sputum smear positivity has long been incorporated into contact investigation screening algorithms [23,30]. However, we do not have enough evidence for the same risk prediction and response measures between contacts of patient with low sputum smear positivity and smear negativity.

Japanese legislation mandates admission and hospitalization of smear positive pulmonary TB patients until three follow-up smear AFB tests are negative following treatment. However, international standards note that the exclusive use of health facility-based directly observed treatment (DOT) may be associated with disadvantages [31].

This study had several limitations. First, the relatively small sample size may have prevented the detection of statistical significance. Second, we were unable to evaluate the demographics or cough frequency in index patients, and we did not have access to associated environmental factors. Environmental factors that would have been useful for analysis include individual contact duration time, information on each patient’s living situation including the level of crowding, and proximity to the index patient, as these may have affected risk of infection. Third, IGRA readouts suffer from within-subject variability, which may have affected the study findings [27]. Lastly, HIV infection status was not described; however, since HIV prevalence is very low, 0.8 cases per 100,000 in the Ibaraki prefecture, the overall effect on this dataset would be limited.

Given the limitations described above, further studies need to be implemented to assess the differences in IGRA positivity between contacts of TB patients and low sputum smear positivity and smear negativity. These studies would include larger sample sizes and evaluate environmental factors present in various settings. Relative risk as well as statistical significance based on differences in positive contact results between groups should be considered for TB response decisions.

## 5. Conclusions

This study did not find significant differences in the number of IGRA-positive contacts of index TB patients with low sputum smear positivity and those of patients with negative smears. Future studies with larger sample sizes and evaluation of environmental factors are necessary to conclusively evaluate the relationship of low smear positivity and smear negativity of TB patients with infection risk of contacts.

## Figures and Tables

**Figure 1 ijerph-16-03713-f001:**
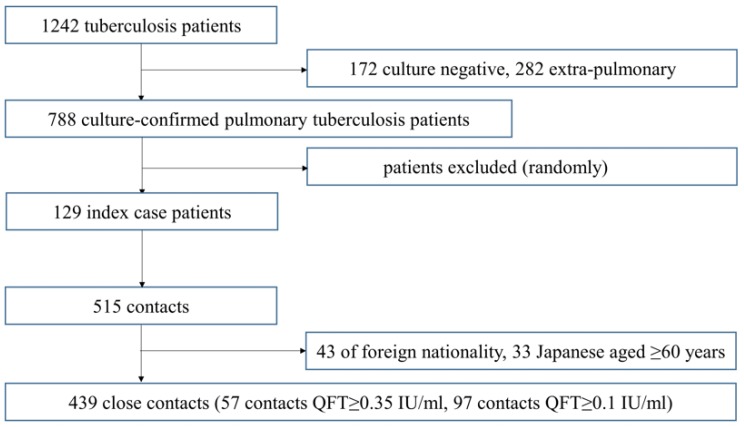
Flow chart of study outline.

**Table 1 ijerph-16-03713-t001:** Characteristics of close contacts.

Variables	Close Contact (*N* = 439)
Age	
0–19	70 (16%)
20–29	74 (17%)
30–39	105 (24%)
40–49	98 (22%)
50–59	92 (21%)
Sex	
Male	220 (50%)
Female	219 (50%)
Cough in index patient	
+	243 (55%)
−	196 (45%)
Cavitation on radiographs of index patient	
+	221 (50%)
−	218 (50%)
Sputum smear AFB of index patient	
−	103 (23%)
Scanty	41 (9%)
1+	115 (26%)
2+	132 (30%)
3+	48 (11%)

**Table 2 ijerph-16-03713-t002:** QuantiFERON^®^-TB Gold In-Tube (QFT) positivity of index tuberculosis (TB) patient close contacts.

Variables	Close Contacts of Index TB Patient			
	QFT ≥ 0.35 IU/mL	QFT < 0.35 IU/mL	Univariate Analysis	Multivariate Analysis
	(%)	(%)	OR (95% CI)	aOR (95% CI)
*N*	57 (13%)	382 (87%)		
Index TB patient				
Cough				
+	41 (17%)	202 (83%)	2.28 (1.2–4.2)	1.80 (0.94–3.5)
−	16 (8%)	180 (92%)	1	1
Cavitation on radiographs				
+	31 (14%)	190 (86%)	1.20 (0.69–2.1)	0.95 (0.49–1.8)
−	26 (12%)	192 (88%)	1	1
Sputum smear				
−	10 (10%)	93 (90%)	1	1
Scanty	3 (7%)	38 (93%)	0.73 (0.19–2.8)	0.68 (0.17–2.8)
1+	11 (10%)	104 (90%)	0.98 (0.40–2.4)	1.12 (0.45–2.8)
2+	16 (12%)	116 (88%)	1.28 (0.56–3.0)	1.20 (0.48–3.0)
3+	17 (35%)	31 (65%)	5.10 (2.1–12.3)	4.96 (1.9–12.9)
Close contacts of index TB patients				
Age				
0–19	11 (16%)	59 (84%)	1	1
20–29	7 (9%)	67 (91%)	0.56 (0.20–1.5)	0.49 (0.17–1.4)
30–39	10 (10%)	95 (90%)	0.56 (0.23–1.4)	0.58 (0.22–1.5)
40–49	15 (15%)	83 (85%)	0.97 (0.42–2.3)	0.91 (0.37–2.2)
50–59	14 (15%)	78 (85%)	0.96 (0.41–2.3)	1.09 (0.44–2.7)
Sex				
Male	30 (14%)	190 (86%)	1.12 (0.64–2.0)	0.97 (0.54–1.7)
Female	27 (12%)	192 (88%)	1	1

OR = odds ratio, aOR = adjusted OR.

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
