# Peer review of "Association of Low Sputum Smear Positivity among Tuberculosis Patients with Interferon-Gamma Release Assay Outcomes of Close Contacts in Japan"

_ijerph, 2019, doi:10.3390/ijerph16193713_

Round 1

Reviewer 1 Report

In this study Ogata et al compare interferon-gamma release assays (IGRA) between contacts of pulmonary TB patients with acid fast bacillus smear positive and with smear negative using QuantiFERON-TB Gold In-Tube (QFT) assays.

In general the study provides a few insights that might be useful to readers in the field. IGRA is a commercially available test that has been highly studied. Several studies regarding the performance of this test exist.

The materials and methods section will benefit by explaining in detail the protocol used for the QFT assay. Similarly the introduction too needs a much more detailed review of the QFT/IGRA assay. Finally it remains up to the authors to further detail and discuss their results. 

Reviewer 2 Report

In this article entitled ″Association of Low Sputum Smear Positivity among Tuberculosis Patients with Interferon-Gamma Release Assay Outcomes of Close Contacts in Japan″, the authors described the association between QTF assay in contact patient and smear positivity in index patients. In my opinion, further explanations are needed to fully understand this study.  

Major comments:

1-line 91-92: The authors used cutoffs of 0.1 and 0.35 IU/mL for QFT positivity according to Japan Anti-Tuberculosis Association guidelines. Unfortunately, these guidelines are not available in English. I believe the authors should describe more extensively the cutoff values and their association with positivity. As most European and American association used cutoff values of ≥ 0.35 IU/mL or borderline values of 0.1-0.7 IU/mL, 0.1IU/mL seems relatively low for positivity and could potentially bias the study due to low detection specificity.

2- It will be relevant for the study to know how many contacts by index patients there are in this study.

Minor comments:

1-line 47: please add a reference

2-line 179: smear-positive pulmonary

Reviewer 3 Report

Ogata el al has made a nice study on the infectivity of TB patients in relation to their smear grade. It is one of the first papers to investigate the difference between the infectivity of patients with low smear positivity and negative smear.

Concerns:

Several parts of the tables are not mentioned in the text: Cough, cavitation, age. Should this be discussed or omitted?

The cut-off value of 0.1 IU/ml is set according to national guidelines in Japan, which I cannot read. Would it be possible to discuss further, why this cut-off is used? I have seen several papers suggesting a borderline interpretation with different values, fx a Swedish study suggesting values between 0.2-0.99 to be considered borderline (Jonsson et al 2017 Plos One). I have never heard it mentioned that 0.1 should be a cut-off for positivity (must give many false positive).

Reviewer 4 Report

This study was aimed to understand the gamma interferon release assays (IGRAs) that was used for identification of latent tuberculosis infection (LTBI), that what happens between close contacts of pulmonary TB patients. While this study did not find significant differences in the number of IGRA-positive contacts between the close contacts, this information is significant and thus suggesting, that IGRAs cannot be used as a tool for understanding the infection with close contacts with patients. While the authors also claim that further evaluations need to be carried out by considering environmental factors as well more sample size. Even though this is a preliminary study this opens up to address the above-mentioned questions and help in evaluating with more sample. Thus, I recommend this work to be published.
